# Experiences Pertaining to Successful Aging in Middle-Aged Women in South Korea

**DOI:** 10.3390/ijerph20196882

**Published:** 2023-10-04

**Authors:** Do-young Lee, Hyun-ju Kim, A-young Jo

**Affiliations:** 1College of Nursing, Changshin University, Gyeongnam 51352, Republic of Korea; shine@cs.ac.kr; 2College of Nursing, Gyeongsang National University, Gyeongnam 52727, Republic of Korea; fatima21@hanmail.net

**Keywords:** female, healthy aging, middle-aged women, qualitative research

## Abstract

This study aimed to analyze and gain an in-depth understanding of the experiences pertaining to successful aging in middle-aged women in South Korea. A sample of 12 middle-aged women, capable of sharing their lived experiences, was divided into three age-based groups: those in their 40s, those in their 50s, and those aged 60–65 years. The collected data were analyzed using Colaizzi’s phenomenological method. Five theme clusters and ten themes emerged. The experiences of successful aging among middle-aged women were categorized as: “Coping with changes in the mind and body”, “Financially stable life”, “Undergoing the aging process with a healthy family”, “Preparations for dying well”, and “Pursuing a meaningful, harmonious life”. These findings highlight the need for programs that prepare middle-aged women to positively accept and enjoy older adulthood by identifying and addressing the factors essential for successful aging and reducing any negative emotions attached to aging and older adulthood.

## 1. Introduction

The relative poverty rate among older adults in South Korea is much higher than that of other member countries of the Organization for Economic Co-operation and Development. Furthermore, by 2030, South Korea is projected to become a super-aged society, whereby older adults will account for more than 20% of the total population. Moreover, the extremely low fertility rate and the resultant reduction in the economically active population might further exacerbate the issues related to aging in South Korea [1]. Particularly, South Korean women, as compared to men, tend to have lower social security contributions [2] and higher average life expectancy [3]. This is likely to result in extended periods of poverty among older women with serious psychological and social issues. Furthermore, South Korea is witnessing a shift in attitudes toward caregiving; the notion that families are responsible for the care of aging members is being replaced by the idea that individuals must care for themselves as they age [4]. Therefore, while addressing the issues of older adulthood, one must focus on factors promoting successful aging to enable older adults to live a healthy and productive life.

Aging is not limited to older adults. It is a common experience that transcends individual differences, impacting people universally. It is a process of psychological and mental maturation and growth [5]. In this regard, successful aging does not mean “no aging”; rather, it focuses on “aging well” [6]. To achieve this, it is crucial that individuals discover and acknowledge the meaning of aging and the contributing factors for successful aging during middle adulthood and prepare accordingly [7]. A study that implemented a program that promoted successful aging in older adults reported an improved successful aging score after the program [8]. The quality of life in older adulthood depends on how individuals spend their middle adulthood; therefore, it is important that middle-aged women be prepared to accept older adulthood.

Middle-aged women undergo an array of physical changes as they near the climacteric phase, including dysmenorrhea and menopause due to loss of ovarian follicular function and hormonal changes [9]. Additionally, they become vulnerable to psychological issues such as depression, anxiety, and feelings of worthlessness as they transition from a productive to a stagnant role. They also face an increased risk of chronic diseases due to accelerated aging [10]. Further, they have inherent health risks owing to deep-rooted societal issues such as parenting, being the primary homemaker, and working in male-centric organizations [11]. Therefore, gaining an understanding of the lives of middle-aged women to promote successful aging has significant implications for providing holistic nursing care.

In this context, this study aims to analyze and gain an in-depth understanding of the experiences related to aging in middle-aged women based on their qualitative narratives. The key study question, therefore, is, “What is the process of successful aging as experienced by middle-aged women?”

## 2. Materials and Methods

### 2.1. Study Design

To shed light on the essence of successful aging in middle-aged women, we conducted interviews with them and analyzed the data using Colaizzi’s phenomenological method [12], which helps discover and accurately describe stated meanings. The study was conducted in adherence with the Standards for Reporting Qualitative Research guidelines (https://www.equator-network.org/reporting-guidelines/srqr, accessed on 15 July 2023).

### 2.2. Participant Selection and Demographics

Middle-aged women (40–65 years) residing in G province were recruited via snowball sampling. Those who were found capable of adequately describing their experiences were selected. A total of four women in their 40s, five in their 50s, and three aged between 60 and 65 years were recruited. Regarding the participants’ educational level, four had completed high school, seven had a university graduate degree, and one provided no response. Economic status was low (n = 2) and middle (n = 10). Six women lived with their children, and six did not. Ten of them followed a religion, while two did not. Ten of them were homeowners, while two were not. Additionally, 11 were employed, and one was not (Table 1).

### 2.3. Data Collection

The data were collected for three months (July–September 2022). The term “middle-aged women” encompasses a broad age range (40–65 years), and the experiences of successful aging would differ across age groups; that is, those in their 40s, those in their 50s, and those aged 60–65 years. There was an age difference between the ages of 40 and 65, so the focus interviews were organized into similar age groups so that they could express their rich experiences together. Hence, focus group interviews were conducted separately for the three age groups. Each interview lasted nearly 1 h and was conducted in a quiet room. Non-structured and open-ended questions were used, and the interviews were recorded upon obtaining consent from the participants. The recordings were transcribed ad verbatim. The non-structured interview questions were: “What is successful aging according to middle-aged women?”, “What are middle-aged women’s experiences of successful aging?”, and “How do middle-aged women prepare for successful aging?”

### 2.4. Data Analysis

We utilized a qualitative approach, following the six-stage phenomenological analysis method proposed by Colaizzi [12]. In stage 1, in-depth interviews were conducted to gain a comprehensive understanding of the participants’ experiences, and we confirmed that our understanding of their statements was correct. The recorded interviews were thoroughly reviewed and transcribed verbatim to accurately capture their statements. In stage 2, we carefully examined the transcripts to identify significant statements that were repeated or emphasized. In stage 3, meaningful statements were expressed in a more general or abstract form. In stage 4, themes, theme clusters, and categories were established based on the organized meanings, ensuring consistency and coherence in the abstraction process. To validate the phenomenological analysis, we sought the insights of two nursing professors experienced in phenomenological research and incorporated their feedback. In stage 5, the theme clusters were named using complete and clear statements that represented the essence of participants’ lived phenomenological experiences. Finally, in stage 6, the essential structure was confirmed with the participants to ensure that it aligned with their lived experiences, and the common factors of the identified phenomena were integrated.

### 2.5. Rigor

To ensure the rigor of this study, we sought to enhance its credibility and validity based on Lincoln and Guba’s [13] four criteria for qualitative research: credibility, transferability, dependability, and confirmability. To achieve credibility, we shared the analysis results with the participants to ensure that the results aligned with their experiences. To ensure transferability, data collection continued until saturation, where no new information emerged from the participants. For dependability, two experienced professors in qualitative research reviewed and evaluated the research findings. Finally, to maintain confirmability, we engaged in discussions to address any potential biases or preconceptions that might have influenced the study.

Interviews for data collection were conducted in Korean, and the content validity of the data was verified by the subjects. Afterward, it was written in English, and the contents were confirmed by a qualitative researcher who is fluent in Korean and English.

### 2.6. Ethics Statement

This study was approved by the Institutional Review Board (IRB) of the Changwon Fatima Hospital (IRB No. CFH-2022-05). Informed consent was obtained from the participants.

## 3. Results

The meaningful statements derived from the successful aging experiences of middle-aged women were analyzed using Colaizzi’s [12] method, and 10 themes emerged. These were then clustered into five theme clusters: “Coping with changes in the mind and body”, “Financially stable life”, “Undergoing the aging process with a healthy family”, “Preparations for dying well”, and “Pursuing a meaningful, harmonious life” (Table 2).

### 3.1. Theme Cluster 1 – Coping with Changes in the Mind and Body

To middle-aged women, successful aging meant maintaining good physical and mental health throughout older adulthood. Theme cluster 1 encompassed their experiences of gracefully embracing the changes in their mind and body that accompany aging, adeptly managing health concerns, focusing on themselves, and fearlessly embarking on a journey of self-discovery.

#### 3.1.1. Theme 1-1: Accepting the Changes of the Body and Mind

Middle-aged women accepted the physical changes resulting from aging as a natural process and strove to focus on their inner selves rather than their appearance.

“You know, accepting this, accepting that you are getting older from your heart. When you are in your 50s, you should have wrinkles that match your age, and in your 60s, you should have wrinkles that are appropriate for that age. It is not right that a woman in her 60 wants to look like she is 40”.(Participant 5)

#### 3.1.2. Theme 1-2: Coping with Physical Aging and Striving to Overcome It

Middle-aged women incorporated exercise into their daily lives to gain energy and health and cope with age-related physical changes. They viewed successful aging as being treated with dignity and respect as they grew older, and they valued how they were perceived by others. To achieve this, they focused on consuming nutritious food and continued to seek cosmetic care to maintain a youthful appearance.

“I have been working out for a month now. I have set my rules to exercise at least 30 min a day and have been following it. I lost weight and I do not feel tired anymore”.(Participant 7)

“I had so much pain from frozen shoulders, but I went to the hospital and got treatment, and it is all resolved now. You know, getting information, and doing things to resolve the issues to restore health (omitted). I eat only natural food”.(Participant 8)

“There is a host of good information on the internet. I searched online and now I take things like collagen and protein every day to care for myself (omitted). But no matter how healthy you are inside, if you do not look good, you do not get respected”.(Participant 9)

#### 3.1.3. Theme 1-3: Solely Focusing on and Discovering Self

Middle-aged women overcame mental health issues by enjoying their hobbies. To experience successful aging, they took time to focus and reflect on themselves.

“I think you can call ‘finding and pursuing whatever you can do’ and ‘whatever you like to do’ a success”.(Participant 3)

“I just felt so suffocated. I would just run out in the middle of the night. The final diagnosis was a mental health issue (omitted). So I began to learn Changgo. Since learning Changgo, this all got resolved”.(Participant 4)

“I mean, learning an instrument, journaling, or growing plants, are those things not healthy hobbies? Now is the time to look after myself (omitted) I have a high self-esteem, so is that not a kind of success?”(Participant 11)

### 3.2. Theme Cluster 2 – Financially Stable Life

For middle-aged women, successful aging meant being financially stable and accumulating enough wealth that would allow them to pursue anything they wanted to do in their lives. Theme cluster 2 encompassed their experiences of wishing to achieve a financially stable life.

#### 3.2.1. Theme 2-1: Wishing to Continue to Be Economically Active

Middle-aged women desired to be economically active for various reasons, such as “feeling anxious about losing income”, “feeling of getting older more quickly when not working”, and “believing that a moderate level of stress is beneficial”.

“It is better to stay economically active until I receive my pension. I am worried about spending all my savings, so I am trying to work a little”.(Participant 5)

“Without work, people seem to age faster, not just their face but also their thoughts. When you work, you meet people outside, so your social connections expand, and I think that would help delay aging emotionally”.(Participant 6)

“Even when I started working at this age, I found a job that truly appreciates me (omitted). I am older, but I ended up finding a job that suits my physical abilities”.(Participant 8)

“You know, a moderate level of stress is ok. Even if you are stressed a little, you know how to overcome it, so I think it is better to go out and work instead of staying home”.(Participant 10)

#### 3.2.2. Theme 2-2: Wishing for Financial Stability

Middle-aged women wished to be financially stable when they were older, so that they would be able to have the luxury of doing things they wished to do without financial constraints.

“Not wishing to be super rich, but you know, having enough to do things that you want to do…”(Participant 5)

“You need to be financially stable, because that shows on your face, and when you look stable, people around you are more relaxed”.(Participant 6)

### 3.3. Theme Cluster 3 – Undergoing the Aging Process with a Healthy Family

Middle-aged women perceived successful aging as rearing children such that the children are successful in their careers and enjoying a good relationship with their spouse. Theme cluster 3 encompassed their experiences of making sacrifices for the success of their children and wishing to age well with their spouse as their life companion.

#### 3.3.1. Theme 3-1: Providing Maximum Support for Child’s Success

Middle-aged women sacrificed their needs for the success of their children, helped them work toward good jobs, imparted their life’s wisdom, and engaged in babysitting their grandchildren. These women were also concerned about their adult children who were not getting married.

“As the children grow up, parents need to say good things for them and help them. We are nearing death, so we need to help them become independent. Not just financially independent, you know…based on what I have experienced in life, something that will give them strength even after we pass away”.(Participant 8)

“I have two daughters, and when I was young, I gave my all to my children. I gave them rides to everything. And they passed the civil service exam in Seoul just within three years. I babysat my eldest daughter’s kids a lot. Now they are all well off. You do not have much to worry when your children are well off”.(Participant 9)

“These days, what pains me the most is my daughter is over 35 already, but she does not want to get married. It’s something that constantly weighs on my heart. So, even if I think I am going through successful aging, I just keep thinking that it might not be successful aging. It’s something that always lingers in my heart. Moreover, since we have only one daughter, I worry that she might feel very lonely on this Earth after we are gone. There will be many challenges ahead, and if she had siblings, they could share their pains with each other, but that’s not the case here. Figuring out how to handle this is indeed a significant concern”.(Participant 11)

“I told them to learn skills, and that landed them jobs at LG. So, I have two sons, and both work for big companies, so I think I am more relaxed in my mind than other people”.(Participant 12)

#### 3.3.2. Theme 3-2: Viewing Spouse as a Companion and Wishing to Age Successfully Together

Middle-aged women considered their spouses their companions, on whom they could depend until the end, even after their adult children moved out of their homes. They wished to grow old together, successfully.

“I learned that having a good relationship with your spouse is important only after I got old. He was like an arch enemy to me sometimes, but now that my children have moved out, they are in the backseat. Even if I have money and work, I feel like a good relationship with your spouse who can support you is the most important thing for successful aging”.(Participant 5)

“I just kept telling my husband, get the barista certification, get the level 1 chef certification, learn baking. At first, he did not even know how to cook rice. But now he is good. Would that not contribute to preparing for successful aging? You know, you do not know who is going to go first”.(Participant 11)

### 3.4. Theme Cluster 4 – Preparations for Dying Well

Middle-aged women believed that preparing to die well also constitutes successful aging. Theme cluster 4 encompassed their experiences of reflecting on a graceful death and preparing to die well.

#### 3.4.1. Theme 4-1: Reflecting on Death

Middle-aged women forgave and emptied their minds while facing the deaths of their family members, and they wished that they would be able to die as they had prepared. The women also mentioned the need for preparation and education for a good death.

“When my father was in the ICU, I thought, ‘ok, this is what dying means’. You can forgive everything upon death. I really emptied my mind since then. I just wish that for now, no one gets sick and everyone is happy. You know, the health of your family is the most important thing”.(Participant 2)

“What I wish is that when I am nearing my death, I get to live for three days. I want to see the faces of my children for the last time and say good bye, and if I have money, I can tell them where the money is. You know, I think you will need time for that kind of stuff. I just pray for that. Just three days”.(Participant 9)

“Education about good death is necessary. You know, that includes things like advance directives and mindset, and things like that. So, education about good death should be offered to middle-aged women and students too (omitted). When you hear about death, you think of things like a beautiful end, (of) my life”.(Participant 11)

#### 3.4.2. Theme 4-2: Preparing to Die Well

In the 60–65 age group, middle-aged women wished to avoid life-sustaining treatment and have a peaceful death when the time for the decision came. Some participants had abundant knowledge about advance directives, and some had filed their advance directives.

“When you are above 65, you have to go there and sign that when you are healthy. It is a must (omitted). It is good for yourself, for your kids, and for the country”.(Participant 9)

“When you go to the National Health Insurance Service and file the advance directive, there are six categories for which you indicate your preference, such as ‘I do not want mechanical ventilation, I do not want CPR… (omitted)’. When you sign it, you can get a national hospital federation registration card. If you have that, they will not administer life-sustaining treatment to you when you collapse”.(Participant 11)

### 3.5. Theme Cluster 5 – Pursuing a Meaningful, Harmonious Life

Middle-aged women viewed successful aging as having peace of mind and living their lives with meaningful relationships with others. Theme cluster 5 encompassed their experiences of emptying their minds, spending time with their close friends, or volunteering to discover their meaning in life.

#### 3.5.1. Theme 5-1: Finding Peace of Mind

Middle-aged women strove to be satisfied and grateful for their current lives and age as respected adults. They prayed every day to attain mental and spiritual maturity and improve their resilience so that they could overcome their past wounds.

“It is good to be generous to other people. It is not like you are going to take everything (when you die). It is good to have peace of mind (omitted). I am a religious person, so for the sake of spirituality, I wake up early in the morning and pray (omitted) and I just try to be grateful. Do not want more things and just (be grateful for) what I have”.(Participant 7)

“No matter how hurt you are, it is important to overcome your wounds and move on, and even if you come across difficulties and your children act like jerks, it is important to think positively and just get over it. (That way), you can live a healthier life; I think resilience is really important”.(Participant 8)

#### 3.5.2. Theme 5-2: Pursuing a Rewarding, Altruistic Life

Middle-aged women wanted to live a rewarding life by being generous to other people and volunteering. By wielding a good influence on other people and bringing about positive changes, they wished to gain peace of mind and make life more meaningful.

“I want to volunteer. I want to be generous and giving… I feel like things that help others also constitute successful aging”.(Participant 7)

“I volunteer at religious facilities, and I can talk with other people there. I think that helps me relax and makes me healthier. I am more relaxed mentally, and I think I try to engage in such meaningful activities. I try to be good to other people and I wish that there are activities where I can talk to people so that they can change in a positive way”.(Participant 8)

#### 3.5.3. Theme 5-3: Getting along with Others in Social Relationships

Middle-aged women continued to engage in social activities and strove to maintain their interpersonal relationships. Instead of staying home all day, they engaged in activities outside of their homes to obtain positive energy and strength. Participants in the 60–65 age group placed more value on time spent with friends, relying on and helping each other.

“I try to go out with my husband as much as possible. I feel great when I go out. It is a healing experience, and there is always something I get out of it. Many of the times, I get motivated to live my life to the fullest”.(Participant 7)

“I go out as soon as the day breaks. They say that your brain ages quickly if you only stay at home. You know, because your thoughts are constrained. Your thoughts expand if you meet friends, chat, laugh, even talk about other people if you want to. I think that makes you healthy. So, if you have one or two friends who would come running if you are depressed and someone you can meet anytime to grab some food, that is a real success”.(Participant 9)

“You do not go outside when you are depressed, you know. If you do not connect with other people and bond with them, you are bound to be depressed and make extreme choices. (omitted) When I’m going through tough times, having just one friend, the kind of friend I can go to their house and stay for a month, that’s success for me”.(Participant 12)

## 4. Discussion

The study aimed to analyze and gain an in-depth understanding of the experiences and perceptions of successful aging among middle-aged women. The theme clusters identified are discussed below.

First, middle-aged women’s experiences of successful aging involved coping with changes in the mind and body. Middle-aged women sought to cope with and overcome aging by maintaining their physical health through exercise, healthy diet, and health-related information. They also sought to maintain their mental health by unreservedly investing in themselves and actively challenging themselves. Physical health is the first component of successful aging identified by Rowe and Kahn [7], and the women in our study strove to maintain their physical health by maintaining functional independence in daily life, engaging in exercise, making good diet choices, quitting smoking, and modifying their lifestyle. Such efforts to maintain effective physical functions and overcome physical aging are considered essential for successful aging not only among older adults but also among middle-aged women [14]. Lee and Yang [15] noted that because of improved education, economic status, social standing, and changing values compared to the past, middle-aged women in the present time have become less reliant on others and more focused on themselves, pursuing productive self-worth. However, it is important for these women to reflect on their lives to adapt to emotional changes caused by despair when thinking about their past lives, feeling as if their life as a woman has come to an end, or adjusting to their adult child moving out and spouse retiring [14]. In the present study, middle-aged women considered that moving away from their selfless, family-centered lives and instead challenging themselves by pursuing enjoyable hobbies and dedicating time to focus on themselves are valuable experiences contributing to successful aging.

Second, middle-aged women’s experiences of successful aging involved a financially stable life. Middle-aged women desire to be financially stable as they enter older adulthood and work hard to achieve this. Being financially stable in older adulthood reduces their stress and opens the doors to diverse activities, contributing to their perception of successful aging. This, in turn, is directly linked to maintaining health, engaging in recreational activities, and meeting one’s self-realization needs [14]. Lee et al. [16] reported that the degree of successful aging is higher among late middle-aged women with increasing monthly income. In the past, women pursued financial stability so as to not be dependent on their children or other family members. However, middle-aged women today value financial power, as it enables them to enjoy their desired activities without constraints in the later days of their lives [15]. Hence, it indicates that middle-aged women do not perceive financial stability as the core essence of successful aging but rather as a means to achieve other facets of successful aging, such as self-realization.

Third, middle-aged women’s experiences of successful aging involved aging with a healthy family. Middle-aged women gladly sacrificed themselves for the success of their children, and they wished to be supported by their spouses and successfully age with them. Older adults in South Korea belong to a generation that endured adversities such as starvation and marginalization during the Japanese colonial rule, the Korean War, and industrialization. As a result, they tend to value collective values and prioritize their family’s well-being over their own happiness [17]. Consequently, children’s success and prosperity are still major areas of focus and interest among middle-aged and older women in South Korea [15]. Seong [18] reported that middle-aged women may be willing to sacrifice their own health, interpersonal relationships, and hobbies for the sake of their children and that these women require a positive support system involving their spouse to age successfully. Seeking stability and love in a marital relationship is a natural aspiration [19], and a strong marital bond evolves through years of mutual interaction [15]. Despite the evolving trend of aging women becoming less dependent on their families, the health and happiness of family were identified as essential factors in successful aging.

Fourth, middle-aged women’s experiences of successful aging involved preparations for dying well. Middle-aged women contemplated their own deaths, reflected on a graceful passage into death, and took practical steps to die well. This notion of “dying well” encompasses the idea that the quality of death is an integral part of life’s fulfillment. It involves choosing a peaceful death over life-sustaining treatments, exercising one’s medical and legal rights in medical decisions, and embracing death without fear and anxiety to naturally come to the end of life [20]. The middle-aged women in the study perceived death to be approaching, leading them to initiate preparations and engage in self-reflection about the end of life. Some participants even took the step of filing their advance directives. Kim and Lee [20] have shown that older adults exhibiting a higher level of acceptance and preparation for death demonstrate higher psychological well-being. Thus, encouraging middle-aged adults to reflect on death and prepare to die well can provide a foundation for successful aging later in life.

Fifth, the participants’ experiences of successful aging involved pursuing a meaningful, harmonious life. Middle-aged women sought inner peace by cultivating their minds, fostering relationships with significant others, and taking steps to live a more meaningful, altruistic life. Seong [18] revealed that middle-aged women perceived successful aging as involving emotional stability and becoming a person who is trusted by those around them. Lee et al. [21] showed that women expected to deepen their thoughts and gain insight as they age, while also believing that they should not burden others and should act their age. In our study, middle-aged women viewed the process of successful aging as aspiring for spiritual maturity through continuous emotional self-regulation and embracing the ability to be generous and giving. Lee and Lee [22] demonstrated that older women participated in volunteer activities for leisure, self-development, recognition, affiliation, and pursuit of the greater good, and their satisfaction with volunteering activities was positively associated with successful aging. This supports Rowe and Kahn’s [7] theory of successful aging, which emphasizes having high levels of psychological and social functioning. The significance of friendships, as emphasized by all three women interviewed in their 60s, aligns with prior research conducted by Lee and Han [23]. It is presumably because people perceive the need for new relationships when their adult child moves out or their spouse passes away. Lee and Han [23] reported that happiness in older women in their 50s and 60s increases when they have many friends in proximity, and they experience emotionally enriched interactions with close friends, actively listening and confiding in them. These results suggest that continuously interacting with and receiving support from others in one’s social network, that is, having good relationships with others, is an important factor for successful aging.

## 5. Conclusions

This study employed Colaizzi’s phenomenological analysis method to comprehensively explore the essence and significance of successful aging experiences among middle-aged women [12]. As a result, successful aging experiences of middle-aged women were identified across physical, mental, financial, social, spiritual, and reflective dimensions.

Middle-aged women are at a physically, psychosocially, and financially vulnerable period in life, and helping them prepare for successful aging before they enter older adulthood is crucial for public health nursing professionals [24]. Hence, it is important to understand the experiences that contribute toward successful aging among middle-aged women, defined as women aged 40–65 years. Such findings will be useful as foundational data for developing methods and strategies to promote practical preparations for successful aging among middle-aged women.

One key significance of this study is that it offers rich insights that can inform the development of programs that promote successful aging in middle-aged women. It highlights those perceptions and experiences of successful aging that have a crucial impact on the quality of life in relatively longer older adulthood. The findings of this study highlight the need for more diverse research on this topic and programs that help middle-aged women prepare for older adulthood. However, as we included middle-aged women of only a particular region in the Republic of Korea, we recommend that future studies explore the experiences of successful aging in other cultures as well as develop successful aging intervention programs for middle-aged adults based on such findings.

## Figures and Tables

**Table 1 ijerph-20-06882-t001:** Characteristics of participants (n = 12).

No.	Age	Educational Level	Marriage Status	Spousal Status	Children and Residence Status	Religious	Occupations
1	57	University graduate	Yes	Yes	Live together	Haven	Haven
2	63	University graduate	Yes	Yes	Do not live together	Haven	Haven
3	60	No response	Yes	Yes	Do not live together	Haven	Haven
4	63	High school	No	Divorced	None	None	Haven
5	57	University graduate	Yes	Yes	Do not live together	Haven	Haven
6	55	University graduate	Yes	Yes	Do not live together	Haven	Haven
7	53	University graduate	Yes	Yes	Live together	Haven	Haven
8	53	University graduate	Yes	Yes	Live together	Haven	Haven
9	49	High school	Yes	Yes	Live together	Haven	Haven
10	47	University graduate	Yes	Yes	Live together	Haven	Haven
11	49	High school	Yes	Widowed	Live together	None	Haven
12	45	High school	Yes	Yes	Live together	Haven	Haven

**Table 2 ijerph-20-06882-t002:** Theme clusters and themes.

1. Coping with changes in the mind and body	1-1. Accepting the changes of the body and mind
	1-2: Coping with physical aging and striving to overcome it
	1-3. Solely focusing on self and discovering self
2. Financially stable life	2-1. Wishing to continue to be economically active
	2-2: Wishing for financial stability
3. Undergoing the aging process with a healthy family	3-1. Providing maximum support for child’s success
	3-2. Viewing spouse as a companion and wishing to age successfully together
4. Preparations for dying well	4-1. Reflecting on death
	4-2. Preparing to die well
5. Pursuing a meaningful, harmonious life	5-1. Finding peace of mind
	5-2. Pursuing a rewarding, altruistic life
	5-3. Getting along with others in social relationships

## Data Availability

The data are contained within the article.

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
