# Peer review of "Experiences Pertaining to Successful Aging in Middle-Aged Women in South Korea"

_ijerph, 2023, doi:10.3390/ijerph20196882_

Round 1

Reviewer 1 Report

Beginning of the topic of introduction is related to South Korean society, it is not world wide topic. It shall be better to focus on the topic after general problem has presented.  Possibly, the second paragraph of the introduction is more general than the first one. If the author want to focus on South Korean society, it must be expressed within the title and abstract.

The reviewer could not follow why the author mentioned "The attitude toward caregiving in Korea is shifting toward the belief that "individuals" rather than "family", are responsible for their own support." The reviewer felt there was a big leap from the sentences before. 

The table 1 shall be revised. What was the difference between "marriage status" and "Spousal status"? It seems to be simillar meaning. There were no economic status and homeowers, which were written in the manuscript. The number of the chilrdren and residence status seems to be wrong. Seven live together and 4 don't live together and 1 does not have is read from the table. And all the participants have occupations. What occupations did the one who was not employed have? Was the information of religion needed? If so, which religion?

Section 2.3, was the information of the treining and preparation needed? The reviewer did not agree with the significance of writting it in the manuscript.

L85-86, "a broad age range" Why the author used the term "broad"? And what was the intent to describe "the experiences of successful aging would differ across age groups."? In this sentence, what was the meaning "age groups"? 

L87 "focus group inteviews were conducted separately for the three age groups" What does this mean? What was the "three age groups"? The inteview were conducted in not each individual but group? Why? 

Did the interview perform in English? or Mother tongue? If latter, the nuance of the description of the wording by the speakers were suspicious to be distorted by the authors. Is it possible to comment the probability of it in the discussion or limitation?

The results were comprehensible, but it seems to be too long. If possible, the description of total verbs shall be summarize elsewhere.

Five themes were reasonable, but these themes were easy to evolve even if there were no interviews. Is it possible to set priorities of these five themes not merely nominate them? At least, just nominating five themes separately seems to be not enough.

L376- The author mentioned about the difference between 60s and 40-50s, in the first time. Most discussion, the authors describe middle-age group  together, but why the author used comparison between 60s and 40-50s in this context? If possible, please reconsider the age difference in all the other themes. 

Was the first paragraph of the conclusions needed? It does not derive from the results of this study, the reviewer guess. 

The reviewer could not point out the quality of English Language.

Author Response

Beginning of the topic of introduction is related to South Korean society, it is not world wide topic. It shall be better to focus on the topic after general problem has presented.  Possibly, the second paragraph of the introduction is more general than the first one. If the author want to focus on South Korean society, it must be expressed within the title and abstract.

  • We have expressed within the title and abstract.

The reviewer could not follow why the author mentioned "The attitude toward caregiving in Korea is shifting toward the belief that "individuals" rather than "family", are responsible for their own support." The reviewer felt there was a big leap from the sentences before. 

  • We have revised the sentences to ensure smoother transitions.

The table 1 shall be revised. What was the difference between "marriage status" and "Spousal status"? It seems to be simillar meaning. There were no economic status and homeowers, which were written in the manuscript. The number of the chilrdren and residence status seems to be wrong. Seven live together and 4 don't live together and 1 does not have is read from the table. And all the participants have occupations. What occupations did the one who was not employed have? Was the information of religion needed? If so, which religion?

  • We have made revisions to the table.

Section 2.3, was the information of the treining and preparation needed? The reviewer did not agree with the significance of writting it in the manuscript.

  • We have removed that content.

L85-86, "a broad age range" Why the author used the term "broad"? And what was the intent to describe "the experiences of successful aging would differ across age groups."? In this sentence, what was the meaning "age groups"? 

  • We have made modifications to that content.

L87 "focus group inteviews were conducted separately for the three age groups" What does this mean? What was the "three age groups"? The inteview were conducted in not each individual but group? Why? 

  • We have added explanations related to that content.

Did the interview perform in English? or Mother tongue? If latter, the nuance of the description of the wording by the speakers were suspicious to be distorted by the authors. Is it possible to comment the probability of it in the discussion or limitation?

  • We have described that the interviews were conducted in Korean and that the content was reviewed by qualitative researchers proficient in both Korean and English.

The results were comprehensible, but it seems to be too long. If possible, the description of total verbs shall be summarize elsewhere.

  • I didn't fully understand the meaning of your statement. Would it be better to rephrase it like this?

1. Coping with changes in the mind and body

1-1. Embracing Changes in Body and Mind

1-2: Coping with Physical Aging and Striving to Overcome It

1-3. Solely Focusing on Self and Self-Discovery

2. Achieving Financial Stability

2-1. Aspiring to Remain Economically Active

2-2: Aspiring for Financial Stability

3. Navigating Aging with a Healthy Family

3-1. Providing Maximum Support for Child's Success

3-2. Viewing Spouse as a Companion and Aiming for Successful Aging Together

4. Preparing for a Good Death

4-1. Contemplating Death

4-2. Preparing for a Peaceful Passing

5. Pursuing a Meaningful and Harmonious Life

5-1. Seeking Inner Peace

5-2 Striving for a Fulfilling and Altruistic Life

5-3. Building Positive Social Relationships

Five themes were reasonable, but these themes were easy to evolve even if there were no interviews. Is it possible to set priorities of these five themes not merely nominate them? At least, just nominating five themes separately seems to be not enough.

  • We believe it would be beneficial to conclude the experience of successful aging as "living a meaningful life" since in phenomenological research, the final thematic grouping often serves as the ultimate conclusion of the study.

L376- The author mentioned about the difference between 60s and 40-50s, in the first time. Most discussion, the authors describe middle-age group  together, but why the author used comparison between 60s and 40-50s in this context? If possible, please reconsider the age difference in all the other themes. 

  • I have removed the section about age group differences and made revisions accordingly.

Was the first paragraph of the conclusions needed? It does not derive from the results of this study, the reviewer guess. 

  • We have made revisions by adding the first paragraph to the conclusion.

Reviewer 2 Report

Congratulations to the authors on a small neat research project. 

Generally it is written well but please note the following points for checking:

Table 1 columns on marriage status, spousal status (what does this mean?), religion and occupation have all gotten scrambled with the term "haven: filling most boxes so this needs fixing and doesnt quite dovetail with the text referring to it.

The rationale for a 25 year age range needs to be made - it is a generational gap of itself so just be clear why this was chosen/allowed.

3.3 - theme title should be "...support for children's success" I think and the quote on line 218-9 seems  a bit disconnected from the theme title so just have a look at that (might be you need a slightly longer section?)

line 317-8 last portion of quote "...If you only have..hard time" seems to be a bit extraneous to the theme? just check and maybe remove or clarify somehow.

line 353 - "..other components.." I think this sentence could elaborate a little more noting the components in summary.

lines 374-79 - it would be good to have a bit more noting and commenting re the age span of participants here.

lines 397-401 claims need to be softened - you only had 3 women in their 60s! So you cant generalise in the manner you have, best you can say is that the three women interviewed in their 60s all noted the significance of friends etc dovetailing with previous research by Lee and Han (one of the authors? if so needs to be noted as such I think)

lines 415-23 need softening - again you only have 12 interviewees so you can only say that they offer rich insights that can certainly inform programs for middle aged women to successfully age, but claiming more than this is overstepping what this research is able to offer on its own.

English is fine - it has a Korean- feel to it. But I prefer to hear the nauncing of English language. Not everyone is British!

Author Response

Table 1 columns on marriage status, spousal status (what does this mean?), religion and occupation have all gotten scrambled with the term "haven: filling most boxes so this needs fixing and doesnt quite dovetail with the text referring to it.

  • We have made revisions to the table.

The rationale for a 25 year age range needs to be made - it is a generational gap of itself so just be clear why this was chosen/allowed.

  • We have made modifications to that content.

3.3 - theme title should be "...support for children's success" I think and the quote on line 218-9 seems  a bit disconnected from the theme title so just have a look at that (might be you need a slightly longer section?)

  • The reason for titling the thematic grouping in 3.3 as "Undergoing the Aging Process with a Healthy Family" is because it encompasses the content related to children in 3.3.1 and the content related to spouses in 3.3.2. We have also extended the section in lines 218-9 to make it a longer segment.

line 317-8 last portion of quote "...If you only have..hard time" seems to be a bit extraneous to the theme? just check and maybe remove or clarify somehow.

  • We have revised this section to clarify its meaning.

line 353 - "..other components.." I think this sentence could elaborate a little more noting the components in summary.

  • We have made more detailed revisions as per your advice.

lines 374-79 - it would be good to have a bit more noting and commenting re the age span of participants here.

  • We have revised that sentence.

lines 397-401 claims need to be softened - you only had 3 women in their 60s! So you cant generalise in the manner you have, best you can say is that the three women interviewed in their 60s all noted the significance of friends etc dovetailing with previous research by Lee and Han (one of the authors? if so needs to be noted as such I think)

  • We have made the revisions as advised. He or she is not one of the authors; he or she is an entirely different person.

lines 415-23 need softening - again you only have 12 interviewees so you can only say that they offer rich insights that can certainly inform programs for middle aged women to successfully age, but claiming more than this is overstepping what this research is able to offer on its own.

We have made the revisions as advised.
